# The Relationship between Medical Staff’s Emotional Labor, Leisure Coping Strategies, Workplace Spirituality, and Organizational Commitment during the COVID-19 Pandemic

**DOI:** 10.3390/ijerph19106186

**Published:** 2022-05-19

**Authors:** Ming-Zhu Yuan, Hsiang-Fei Chen, Cheng-Chia Yang, Tong-Hsien Chow, Chin-Hsien Hsu

**Affiliations:** 1Department of Healthcare Administration, Asia University, Taichung 413305, Taiwan; yuanmingzhu@fzfu.edu.cn (M.-Z.Y.); chengchia@asia.edu.tw (C.-C.Y.); 2Department of Leisure Industry Management, National Chin-Yi University of Technology, Taichung 411030, Taiwan; r91595@yahoo.com.tw; 3Department of Leisure Sport and Health Management, St. John’s University, New Taipei 25135, Taiwan; thchow1122@mail.sju.edu.tw

**Keywords:** emotional labor, leisure coping strategies, workplace spirituality, organizational commitment

## Abstract

Many medical issues have gradually emerged under the severe impact of the COVID-19 pandemic, which has not only changed the medical culture but also tested medical staffs’ response abilities, emotional pressure, sense of identity, and belonging to the organization. The relationships among medical staffs’ emotional labor, leisure coping strategies, workplace spirituality, and organizational commitment during the COVID-19 pandemic are explored in this study. With medical staffs as the research subjects, a questionnaire survey was conducted using convenience sampling; a total of 360 questionnaires were distributed and 330 were returned, for a recovery rate of 91%. There were 300 valid questionnaires after 30 invalid questionnaires were excluded, for an effective recovery rate of 90%. SPSS and AMOS software were used for statistical analysis. According to the research results: (1) emotional labor had a significant effect on workplace spirituality, (2) workplace spirituality had a significant impact on organizational commitment, (3) emotional labor had a negative and significant impact on organizational commitment, (4) emotional labor had a significant impact on leisure coping strategies, and (5) the mediating effect of workplace spirituality between emotional labor and organizational commitment was not significant. Finally, relevant practical suggestions are provided based on the results of this study.

## 1. Introduction

### 1.1. Background

Many medical issues have gradually emerged under the severe impact of the COVID-19 pandemic, which has changed the medical culture and tested the response abilities and emotional pressure of medical staffs. The huge impact of COVID-19 on the psychology of medical staffs has also been revealed by related research [1,2,3]. For example, Sanchez-Gomez et al. [1] used 1549 healthcare workers as the research subjects to explore the impact of emotional intelligence on work performance under the high-pressure influence of the COVID-19 pandemic. The results showed that work engagement plays a mediating effect between emotional intelligence and work performance. On the other hand, Sanchez-Gomez et al. [4] used a post-traumatic stress disorder (PTSD) framework to analyze the fears brought about by COVID-19 and the impact on physical health. In the study, it was obvious that COVID-19 not only affects the psychological aspect of medical staff, but also, the physical aspect should be the focus of attention. Wu et al. [5] pointed out that, in countries with severe pandemic effects, the sad faces of medical staffs crying helplessly when calling for help from the outside world were very touching, and the unfortunate incident of the suicide of a young nurse caring for COVID-19 patients took place in the United Kingdom, which shows that the mental state and work stress index of medical staffs under the pandemic need attention in real time, as well as the formulation of a new care mechanism. The negative emotions and intense pressure on medical staffs during the COVID-19 pandemic can be found from this phenomenon. As their professionalism is irreplaceable, nursing staffs play an important role in pandemic prevention. Chen et al. [6] also held the same view and suggested that, in the face of the COVID-19 pandemic, nursing staffs still have the courage to take on the responsibility and participate in every battle against the virus. Regardless of the delivery of passengers of charter flights, airport quarantine, institutional screening, patient care, community health education guidance, and pandemic prevention and tracking, they all uphold their professional commitment, are not afraid of hardships, and protect the health of the people. However, with the advent of the post-pandemic era, and the need to coexist with the virus, medical staffs face more changes in the medical environment. For example, medical staffs may be discriminated against by their neighbours, relatives, or family members of patients. While testing personnel put on and take off protective clothing in strict accordance with the procedures every day at work, they still worry about contaminating their family members with the virus. Many such invisible pressures may affect the health and safety of medical staffs; therefore, it has become a very important issue to understand how medical staffs deal with their own emotions based on the interests of clients, organizations, and their own interests in a high degree of interactions with patients, family members, and colleagues in the process of work. Therefore, exploring the emotional labor of medical staffs is one of the research focuses of this study.

In addition, under the high work pressure and the impact of the pandemic, medical staffs have begun to think differently about whether to continue to work. Hsu et al. [7] pointed out that, as of November 2020, according to the Taiwan Nursing Workforce Statistics from the Health Information Network of the Ministry of Health and Welfare, after the number of licensed persons over the age of 65 was deducted, the practicing rate of nurse practitioners was only 63.4% (the rate of those who were licensed for nursing and engaged in nursing work), and the cause of the “low practice rate” was the “high turnover (quit) rate” of nursing staffs. Therefore, the high turnover (quit) rate of nursing staffs is an important issue that requires urgent attention, understanding, and solutions. It can be seen that, during the pandemic, although medical staffs want to be involved in work meaningful to society, whether employees still use self-devotion to achieve the process of self-identity in their hearts in today’s medical environment needs further empirical research. Therefore, understanding the workplace spirituality of medical staffs is the second focus of this study.

Since the end point of the pandemic is unknown, some medical staffs have chosen to leave their jobs under severe circumstances, while some nursing staffs continue to maintain a sense of identity and belonging to the organization, because they are properly treated by the organization. For example, Chen [8] reported that “E-Da Hospital said that the efforts and contributions of nursing staffs are worthy of respect. The hospital is doing its best to protect the rights and interests of the nursing staff regarding their workload, vacation, safety, and hygiene. It also hopes the pandemic prevention regulations are abided by thoroughly so that the pandemic will end soon and the nursing staff will get the biggest applause and reward.” Therefore, the third focus of this study is to explore the organizational commitment of medical staffs and to further understand the strength of their identity and participation in an organization.

With the development of the pandemic, the shortage of manpower in hospitals and the opening of additional wards for diagnosed patients are continuing. This situation also means that the workload of medical staffs is increasing, resulting in less free time and no time to engage in leisure activities to adjust the body and mind. Therefore, the fourth research focus of this study is to explore the leisure coping strategies of medical staffs; that is, how to effectively enhance personal self-identity through participation in leisure activities, reduce and manage stress, and seek social support to cope with stress.

It can be seen from the relevant literature that most of the research on medical staffs during the COVID-19 pandemic focused on anxiety and stress [9,10,11,12]. There are also studies that analyzed how to promote the health and well-being of medical staffs during the pandemic [13,14]. In addition, Kahleova et al. [15] explored issues related to the nutrition of medical staffs during the pandemic and analyzed the effect of a plant-based diet on the metabolism and life quality of medical staffs.

Therefore, in the process of work, medical staffs have a high degree of interaction with patients, family members, and colleagues. It is the focus of this study on how to stabilize their emotions, regulate stress through leisure participation, and produce positive results to improve stress, so that they can still achieve the process of self-identification in their hearts through self-devotion in today’s medical environment and present a positive attitude and spontaneous contribution to the organization to maintain a higher centripetal force. The empirical analysis found that emotional labor has a significant impact on workplace spirituality [16,17], workplace spirituality has a significant impact on organizational commitment [18,19,20], emotional labor has a significant impact on organizational commitment [21,22,23], and emotional labor has a significant impact on leisure coping strategies [24,25,26]. On the other hand, there is also relevant research evidence that workplace spirituality has a mediating effect between emotional labor and organizational commitment [20]. As learned from the research mentioned above, the issues of personal behavior and organizational commitment of medical staffs under pressure are rarely discussed. Therefore, this study took medical staffs as the research subjects to carry out an empirical study of the relationships among emotional labor, leisure coping strategies, workplace spirituality, and organizational commitment of medical staffs under the COVID-19 pandemic.

### 1.2. Literature Review

When people face a specific situation or work, they often need to package or modify their emotions to present an image in line with their work. Hochschild [27] pointed out that emotional labor refers to the management of personal commitment to emotions in order to create the facial expressions and body movements that are acceptable to everyone in public, can be sold in exchange for remuneration, and have an exchange value. With medical care work as an example, emotional labor can refer to how medical staffs enjoy the actual inner feelings and maintain positive emotions and expressions as much as possible during their service in the medical field, no matter whether they are facing a patient, family member, or colleague, as this is the only way to make the other party feel better and increase their satisfaction with the service. In other words, this process requires the coordination of one’s own mentality and emotions. As the service industry replaces the traditional agricultural industry as the mainstream of society, the related research on emotional labor is becoming more and more abundant. According to Wharton and Erickson [28], the work of emotional labor has three characteristics: (1) Workers need to contact the public via face-to-face or voice interactions. (2) Workers need to produce appropriate emotional states in front of customers. (3) Emotional labor allows employers to control and restrain employees’ emotional activities to a certain extent through training and supervision. These three characteristics conform to the work characteristics of medical staffs. Morris and Feldman [29] further revised the definition of emotional labor. They mentioned that emotional labor refers to the effort, planning, and control required to express the emotions desired by the organization in the process of interpersonal interactions and took the lead in proposing the measurement dimensions of emotional labor. The five dimensions of emotional labor, as proposed by Lin [30], have been adopted by most related research on emotional labor in Taiwan: (1) basic emotional expression, (2) superficial emotional control, (3) deep emotional camouflage, (4) degree of emotional diversity, and (5) degree of interaction. Chen and Chen [31] pointed out that the overall correct prediction rate of the emotional labor burden scale of Lin [30] was 90.23%, which showed that the scale has a certain level of discrimination.

Participation in leisure activities has long brought about different benefits for people, thereby enhancing personal self-confidence or self-identity. Iwasaki and Mannell [32] pointed out that the so-called leisure coping strategies mean that people will participate in leisure activities when they encounter various life pressures, which can mediate stress and produce positive results to improve stress and maintain health. Coleman and Iso-Ahola [33] were the first researchers to regard leisure as a stress coping method, and in 1993, they proposed that, in the process of leisure participation, individuals can obtain social support and improve their sense of personal decision-making freedom. Among them, the social support of friends during leisure participation can help individuals adjust their life pressure, leisure participation can enable individuals to have successful experiences, and individuals will enhance their sense of personal decision-making freedom by trusting themselves. In this way, individuals can face and reassess their stress and maintain their physical and mental health through these two leisure coping approaches. It can be seen that scholars have continuously considered the effects of leisure on people. In this study, in order to prevent the negative effects of stress, the importance of using leisure participation as a channel for medical staffs to cope with stress also makes leisure one of the important methods for medical staffs to face or manage stress. Iwasaki and Mannell [18] divided leisure coping strategies into three types, according to the individual’s leisure behavior and perceptions:Leisure companionship

Leisure companionship is a type of social support, which means that an individual develops feelings with others and builds relationships through leisure participation, and these friends will lend a hand to help the individual get through the difficulties and pressure they encounter [34].
2.Leisure palliative coping

This is a stress-coping strategy using escape, meaning the use of leisure participation to escape stress and achieve a soothing effect.
3.Leisure mood enhancement

The use of leisure participation to enhance positive emotions is also an important type of stress coping strategy.

The so-called workplace spirituality refers to a life experience in which an organization promotes employees’ self-transcendence through work procedures. If the organization can provide employees with growth opportunities to develop their own abilities or wisdom, it will meet employees’ needs for self-actualization in their own lives [35]. In this study, workplace spirituality is related to the purpose and meaning of work, as well as the sense of group identity of medical staffs. Especially in the current pandemic, how to properly deal with the balance and conflict between work and life is the focus of workplace spirituality. From this point of view, the workplace spirituality of medical staffs echoes the definition of Ashmos and Duchon [36], who proposed that workplace spirituality has three dimensions, including inner life, meaningful work, and community. In other words, when medical staffs at work feel that the work they are doing is meaningful, they can stay in the medical field for a long time, as they identify with the content of the work. At the same time, if medical staffs are willing to stay in their unit for a long time, it can be said that their work is meaningful and can nourish the individual’s inner life. Hsu et al. [37] further differentiated workplace spirituality into individual-level spiritual perception and organizational-level organizational spirituality. Spirituality perception refers to an individual’s realization of the values and meaning of his or her life through his or her work or workplace experience. It is an act of introspection that generates new thinking beyond ordinary sensory experience through the development of deep connections between the individual and self, others, society, and all things in nature. Organizational spirituality refers to the organization establishing the perception and care of a community, which allows employees to feel the meaning of their work in life through spiritual conversation and listening and overcoming the various pressures and challenges via spiritual learning and growth. Based on this classification, this article argues that workplace spirituality emphasizes interdependent relationships—that is, the inseparability of oneself and the organization and the joint construction of common experience.

Organizational commitment explores an individual’s identification with organizational goals and values and the degree to which they are willing to invest. Robbins [38] considered organizational commitment as a work attitude—that is, the degree to which employees identify with a specific organization and its goals and want to maintain membership in the organization. Cao et al. [39] defined organizational commitment from a psychological state and pointed out that organizational commitment is a psychological state that describes the relationship between employees and an organization and may influence an employee’s decision to continue or terminate their membership in the organization. As research on organizational behavior becomes more and more abundant, the classification of organizational commitment is obvious in the related research. Allen et al. [40] divided organizational commitment into (1) affective commitment, which refers to employees’ attachment behavior and sense of belonging to the organization, and (2) continuance commitment—that is, the perception of the cost of leaving the organization—resulting in a commitment to stay in the organization, where the employee’s measurement of the number of other employment opportunities and the sacrifice required to leave the company are the main sources of the cost, and (3) normative commitment, which refers to the loyalty of employees to the organization and is an obligatory responsibility. Although the classifications of organizational commitment have been made by different scholars, Chen and Lin [41] pointed out that the classification of organizational commitment generally includes two types, the psychological aspect and the exchange aspect. The former is the willingness to devote oneself to the organization from the heart—that is, starting from the point of view of attitude; the latter means that the individual will evaluate the profit and loss and then determine their degree of dedication to the organization by considering the factors that will benefit oneself. In this study, organizational commitment can be described as the positive attitude of medical staffs toward the organization, and the organization actively takes care of medical staffs in order to enable them to contribute to the organization and maintain a higher centripetal force.

It is worth noting that work as a calling has been added by some studies for discussing organizational commitment to further explore how work as a calling affects members’ performance and thinking in the organization [42,43]. That is, work as a calling is regarded as an action process that provides the meaning and purpose of the individual’s work, thereby increasing the individual’s degree of dedication to the organization. The above-mentioned related literature was sorted, the research hypotheses were developed for this article, and empirical data were collected to further explore the relationship between medical staffs’ emotional labor, leisure coping strategies, workplace spirituality, and organizational commitment during the COVID-19 pandemic.

## 2. Methods

### 2.1. Research Structure

This study mainly discussed the relationships among the emotional labor, leisure coping strategies, workplace spirituality, and organizational commitment of medical staffs during the COVID-19 pandemic. The research structure was proposed according to the literature review and research purposes, and the research structure is shown in Figure 1.

### 2.2. Research Hypotheses

**Hypothesis** **1** **(H1).**
*Emotional labor has a significant impact on workplace spirituality.*


**Hypothesis** **2** **(H2).**
*workplace spirituality has a significant impact on organizational commitment.*


**Hypothesis** **3** **(H3).**
*Emotional labor has a significant impact on organizational commitment.*


**Hypothesis** **4** **(H4).**
*Emotional labor has a significant impact on leisure coping strategies.*


**Hypothesis** **5** **(H5).**
*workplace spirituality has a significant mediating effect between emotional labor and organizational commitment.*


### 2.3. Research Subjects

This study enrolled nurses and emergency ambulance technicians as the research subject and employed convenience sampling as the research method. Between 15 January and 15 February 2022, questionnaires were distributed in Taichung Armed Forces General Hospital, Everan Hospital, China Medical University Hospital, Ministry of Health and Welfare Taitung Hospital, Ministry of Health and Welfare Hualien Hospital, Ministry of Health and Welfare Pingtung Hospital, and Cheng Ching Hospital onsite. Links to Google Forms were also provided in social software, so that nurses and emergency ambulance technicians could participate in this study via either method. A total of 360 questionnaires were issued, and 330 questionnaires were collected, for a return rate of 91.67%. After excluding invalid questionnaires and those with an incomplete filling of answers, 300 valid questionnaires were collected, and the effective recovery rate was 91.00%

### 2.4. Research Tools

The content of the questionnaire in this research was compiled mainly by referencing the relevant literature and the modified questionnaires of Tseng [20] and Yang [26]. The questionnaires were distributed to nurses and emergency ambulance technicians. The questionnaire was divided into five parts, with a total of 74 items, including 9 items of personal basic information, 21 items of emotional labor, 16 items of leisure coping strategies, 16 items of workplace spirituality, and 12 items of organizational commitment. A five-point Likert scale was adopted by this study, and each question was given 1–5 points, from “strongly disagree” to “strongly agree”, respectively.

### 2.5. Data Processing and Analysis

After the valid questionnaires were collected and the invalid questionnaires were excluded, the data were archived with SPSS 22.0 statistical software, and the correlation between the variables was analyzed with AMOS 22.0 statistical software.

## 3. Results and Discussion

### 3.1. Sample Characteristics

In this study, there were 300 valid samples. The sample characteristics are shown in Table 1. It can be seen from Table 1 that most subjects were female, and most were 31–40 years old. The marital status of most of them was married, most of them had a university educational level, and the residence of most of them was in the central area. The job title of most was nonexecutive, most had more than 10 years of service, and most of them did not often work overtime either on weekdays or on holidays.

### 3.2. Measurement Model Analysis

This study used a confirmatory factor analysis to test the reliability and validity of the questionnaire, and the items were revised with reference to modification indices (MI) [44]. Items B1, B4, B5, B8, B9, B10, B11, B12, B15, B19, B20, and B21 of the Emotional Labor Scale; Items C1, C2, C7, C11, and C15 of the Leisure Coping Strategies Scale; Items D1, D2, D6, D7, D10, D15, and D16 of the workplace spirituality Scale; and Items F3, F4, F5, F7, F11, and F12 of the Organizational Commitment Scale were deleted in this study.
(1)Test of Convergent Validity

Table 2 is a test of the convergent validity of the Emotional Labor Scale. The test results showed that the values of the factor loading of all dimensions in this study were between 0.77 and 0.88, the composite reliability (C.R.) values were between 0.79 and 0.86, and the average variance extracted (AVE) values were between 0.65 and 0.68, indicating that the Emotional Labor Scale of medical staffs has a convergent validity.

Table 3 is the test of the convergent validity of the Leisure Coping Strategies Scale. The test results showed that the factor loading values of all dimensions in this study ranged from 0.60 to 0.91, the composite reliability (C.R.) values ranged from 0.80 to 0.93, and the values of average variance extracted (AVE) were between 0.59 and 0.77, indicating that the Leisure Coping Strategies Scale for medical staffs has a convergent validity.

Table 4 is a test of the convergent validity of the workplace spirituality Scale. The test results showed that the factor loading values of all dimensions of this study were between 0.72 and 0.9, the composite reliability (C.R.) values were between 0.88 and 0.89, and the average variance extracted (AVE) values were between 0.66 and 0.79, indicating that the workplace spirituality Scale of medical staffs has a convergent validity.

Table 5 is a test of the convergent validity of the Organizational Commitment Scale. The test results showed that the factor loading values of all dimensions of this study were between 0.72 and 0.9, the composite reliability (C.R.) values were between 0.79 and 0.90, and the average variance extracted (AVE) values were between 0.65 and 0.81, indicating that the Organizational Commitment Scale of medical staffs has a convergent validity.
(2)Discriminant Validity

In this study, the confidence interval method was adopted. AMOS was used to provide two confidence interval estimation methods, with one being the bias-corrected percentile method, and the other being the percentile method. Table 6, Table 7, Table 8 and Table 9 list the correlation coefficient values among the facets calculated by Bootstrap. The estimated values all fell within the upper and lower limits of the confidence intervals of the bias-corrected percentile method and the percentile method, indicating that there was good discriminant validity among the facets [45].

Table 7 is the test results of the 95% confidence interval of the Bootstrap correlation coefficient of the Leisure Coping Strategies Scale. The results showed that 1 did not appear in the 95% confidence interval of the bootstrap correlation coefficient for the dimensions of the Leisure Coping Strategies Scale, indicating a good discriminant validity.

Table 8 is the test result of the 95% confidence interval of the Bootstrap correlation coefficient of the workplace spirituality Scale. The results showed that 1 did not appear in the 95% confidence interval of the bootstrap correlation coefficient for the dimensions of the workplace spirituality scale, indicating a good discriminant validity.

Table 9 is the test result of the 95% confidence interval of the Bootstrap correlation coefficient of the Organizational Commitment Scale. The results showed that 1 did not appear in the 95% confidence interval of the bootstrap correlation coefficient for the dimensions of the Organizational Commitment Scale, indicating a good discriminant validity.
(3)Test of Mediating Effect

Cheung [46], Cheung and Lau [47], and Lau and Cheung [48] put forward a specific judgment method for the type of mediating effect: if the 95% confidence interval of the indirect effect value does not include 0, it is significant and represents that there is a mediating effect. Since the *p*-value was not significant, hypothesis 5 was not supported, as shown in Table 10.
(4)Structural Model Analysis

The structural model analysis of this study referred to the opinions of Bagozzi and Yi [49], Benter [50], Wu [51], and Hair et al. [52], and seven indicators of χ2 test, the ratio of χ2 to degrees of freedom, GFI, AGFI, RMSEA, CFI, and PCFI were used to evaluate the overall goodness for fit. Table 11 shows that the corrected χ2 is 1115.51, the ratio of χ2 to degrees of freedom is 2.05, the value of GFI is 0.83, the value of AGFI is 0.80, the value of RMSEA is 0.06, the value of CFI is 0.93, and the value of PCFI is 0.85; thus, this model is acceptable. 

As shown in Figure 2 and Table 12, H1 was established; that is, emotional labor had a significant impact on workplace spirituality. The research results of this study were the same as those of Zou et al. [17], and the possible reason was that medical staffs can make appropriate emotional responses and create a better working atmosphere when they are facing patients, family members, or supervisors, meaning that medical staffs can identify and feel meaningful about their work. H2 was established; that is, workplace spirituality had a significant impact on organizational commitment. The research results of this study were the same as those of Tseng [20], and a possible factor was that, once medical staffs identify with the meaning of their work, they will put the maximum effort into the work, or they will not consider quitting due to the serious nature of the pandemic. Work as a calling has been added by some studies for discussing organizational commitment to further explore how work as a calling affects members’ performances and thinking in the organization [42,43]. That is, work as a calling is regarded as an action process that provides the meaning and purpose of the individual’s work, thereby increasing the individual’s degree of dedication to the organization. H3 was established; that is, emotional labor had a negative impact on organizational commitment. The research results of this study were the same as those of Lee [22] and Kismet and Erkan [21], and the possible reason is that if medical staffs feel there is related assistance in managing themselves and dealing with other people’s emotions, they may show a higher sense of identity and centripetal force to the organization, which helps them to understand that the medical unit or organization can provide support. H4 was established; that is, emotional labor had a significant impact on leisure coping strategies. The research results of this study were the same as those of Yang [26], and a possible reason is that medical staffs can properly express and manage their emotions and sort out negative emotions by participating in leisure activities while maintaining a good mood for working under high pressure.

## 4. Suggestions

Some suggestions are put forward by this article for reference, based on the above research results: (1)For Nursing Staffs

The research results of this study showed that emotional labor had a significant impact on workplace spirituality. Therefore, it is suggested that medical staffs should face their emotions, as both positive and negative emotions are feelings produced by staying in the workplace for a long time. In particular, when nursing staffs are providing care in the medical process, they face not only patients and visitors but also family members; thus, if the management system of the service unit is not perfect, it is important to improve the emotional labor of medical staffs who are providing services under the pressure of multiple aspects. Therefore, proper guidance and learning can help medical staffs to release their emotions. For example, before caring for patients with COVID-19, if medical staffs could understand in advance that the pressure and emotional responses of these patients would be different from those of general patients and consult colleagues or seniors regarding strategies for regulating emotions in the front line, it would have helped medical staffs to face this high-stress situation. In addition to their own emotions, medical staffs need to accept the negative emotions of patients and especially the isolation of patients with COVID-19, which makes them socially separated from others, and they tend to have feelings of abandonment. Therefore, if medical staffs can properly use empathy to care for and listen to the negative emotions of patients and provide medical expertise and medical resources support, it will help patients to relieve their anxiety. This interactive process will not only reduce the impact of emotional labor for nursing staffs, but such dedication to patients will also maintain good interpersonal relationships and establish a stronger support system, which will allow medical staffs to have a high sense of identity in their work, understand the value of life due to their efforts and the meaning of their work, and improve their workplace spirituality. The results of this study showed that emotional labor had a significant impact on leisure coping strategies. Therefore, it is recommended that medical staffs should seek to establish a channel to express stress, reduce work stress, and manage emotions through participation in leisure activities. For example, engaging in socially supported leisure companionship is an approach in which a leisure sports community is established; then, the place engaged in leisure sports can usually become a social place. Such community initiatives can promote social connections and psychological support for medical staffs and help transfer the pressure brought about by negative emotions through investment in leisure sports, so that medical staffs can further obtain the benefits brought about by experiencing leisure sports. These methods all help to stabilize emotions. For example, the New Start Sports Center, which is located on the seventh floor of the Health Management Building of Tai’ an Hospital, includes large aerobics classrooms, small aerobics classrooms, cardio and weight training areas, and flywheel areas, and various courses are provided. If medical staffs have the opportunity to develop the habit of continuous and regular exercise, it will play a role in buffering and regulating negative pressure, thereby helping medical staffs to maintain their physical and psychological health.
(2)For Medical Institutions and Organizations

The results of this study show that emotional labor had a negative impact on organizational commitment, and it reached a significant level. Therefore, it is suggested that medical institutions and organizations should establish an in-hospital care mechanism. Especially, it is inevitable that emotional labor issues will be involved in the process when the first-line medical staffs who bear the brunt use empathy to accompany, care for, and offer consolation to patients and their family members; thus, medical staffs can build trust and harmony between doctors and patients; therefore, it is urgent to establish a care mechanism in hospitals. The COVID-19 Mental Health Guidelines for Employees of Medical Institutions, as released in 2021 by the Centers for Disease Control, Ministry of Health and Welfare, can be regarded as information worthy of reference by medical institutions and organizations, such as establishing a task force of general mental health services or setting up employee mental health services using the institution’s original employee care mechanism or one established under the structure of the pandemic prevention command center in response to the pandemic, thus providing employee consultations through more than one channel. In other words, medical institutions and organizations can help strengthen the mental health of medical staffs and reduce their emotional pressure through such measures. Once medical staffs feel that the medical institutions and organizations are attentive to assisting them in seeking out an appropriate outlet for pressure to reduce emotional labor, medical staffs can increase their emotional attachment to medical institutions and organizations, identify with the organizations, and share their goals and values. The results of this study showed that workplace spirituality had a significant impact on organizational commitment. Therefore, it is recommended that medical institutions and organizations should establish harmonious relationships that cover medical staffs and patients, colleagues, supervisors, and family members. In particular, medical staffs have different psychological needs and emotional issues. For example, the psychological pressure of medical staffs who care for confirmed patients and who are confirmed patients themselves is different. Therefore, medical institutions and organizations should develop psychological support service plans to provide assistance to medical staffs through appropriate professional manpower and prepare graded service contents based on expert advice and relevant experience. Through such measures, medical staffs will feel that medical institutions and organizations can provide strong backing, which allows them to fully devote themselves to medical care work, achieve the identification of themselves and their lives in their own hearts, and ultimately, continue to identify with the medical institutions and organizations they serve.
(3)Recommendations for Future Research

The results of this study indicated that workplace spirituality did not have a significant mediating effect between emotional labor and organizational commitment. However, in general, when the emotional labor of medical staffs is increased, they should be able to improve their own workplace spirituality to achieve positive improvements in organizational commitment. While this article only discussed workplace spirituality related to “workplace work”, there are also more and more related research on “spiritual health” as related to well-being and happiness. For example, Li et al. [53] conducted research on the spiritual health and job satisfaction of nursing staffs, and the results of their research showed the relationship between the themes of different spiritual research and the personal and career lives of medical staffs. In addition, the research direction related to learning and growth, as well as one’s own attitude toward life, is another topic of spirituality-related research. For example, Lo and Shiau [54] emphasized the importance of nurturing spiritual life for nursing staffs and explored building up the value of humanity, promoting spiritual growth, driving life renewal, implementing whole-person care, and other topics with nursing staffs as the research subjects, which indicates that there is diverse spiritual-related research. Thus, future related research can include broader spiritual issues beyond the “workplace work” when discussing mediation effects, and semi-structured interviews can be used to obtain more nursing staff’s inner processes. The integration of such materials will help to understand the practical significance of spirituality-related issues for enterprise organization management.

On the other hand, the theoretical framework of emotional labor, leisure coping strategies, workplace spirituality, and organizational commitment constructed in this study has been verified through empirical research. In the application of the theoretical framework in the future, it is suggested that this theoretical framework can be applied to other working groups of high emotional labor services, such as flight attendants, caregivers, department store counter staffs, and first-line customer service staffs, to understand whether different degrees of physical and psychological impacts of the same high emotional labor work groups in the face of the COVID-19 pandemic have an impact on leisure coping strategies, workplace spirituality, and organizational commitment. 

Finally, the focus of the public’s attention is often on the footprints of confirmed cases, the transmission chain, vaccine administration, community transmission, and pandemic prevention policies, though the burden of maintaining the normal operation of the entire medical system and implementing pandemic prevention policies often falls on medical staffs. However, the pressure on medical staffs and the issues of emotional adjustment are often easily ignored by the public, which, in turn, leads to more medical staffs quitting their jobs. Therefore, this study takes medical staffs as an example to describe the emotional labor faced by medical staffs currently in the COVID-19 pandemic through empirical research. The article also proposes appropriate leisure coping strategies to enhance medical staffs’ sense of work purpose and meaning, as well as group identity, through leisure coping to improve the organizational commitment of medical staffs in order for them to continue to make good use of their professionalism in the medical system. Thus, this study has a certain degree of practical contribution.

## 5. Conclusions

It could be seen through empirical analysis that the emotional labor of medical staffs had a significant impact on workplace spirituality during the COVID-19 pandemic. In addition, the research results also pointed out that the workplace spirituality of medical staffs had a significant impact on organizational commitment, the emotional labor of medical staffs had a negative and significant impact on organizational commitment, and the emotional labor of medical staffs had a significant impact on leisure coping strategies. Finally, this study found that the mediating effect of medical staffs’ workplace spirituality between emotional labor and organizational commitment was not significant. Future related research can include broader spiritual issues beyond “workplace work” when discussing mediation effects, and semi-structured interviews can be used to obtain more nursing staffs’ inner processes. The integration of such materials will help to understand the practical significance of spirituality-related issues for enterprise organization management.

## Figures and Tables

**Figure 1 ijerph-19-06186-f001:**
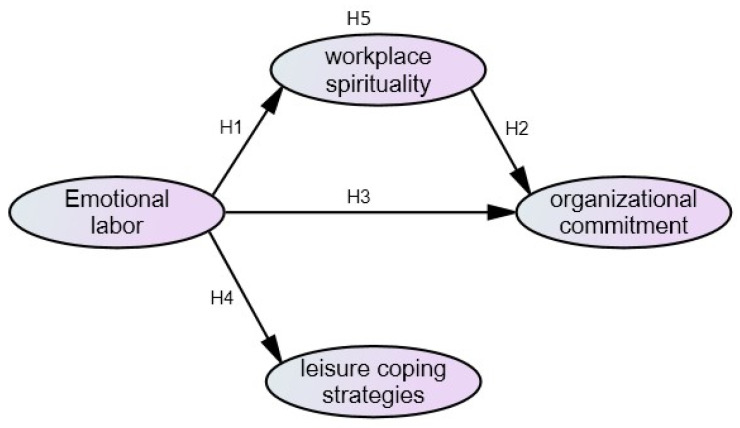
Conceptual framework.

**Figure 2 ijerph-19-06186-f002:**
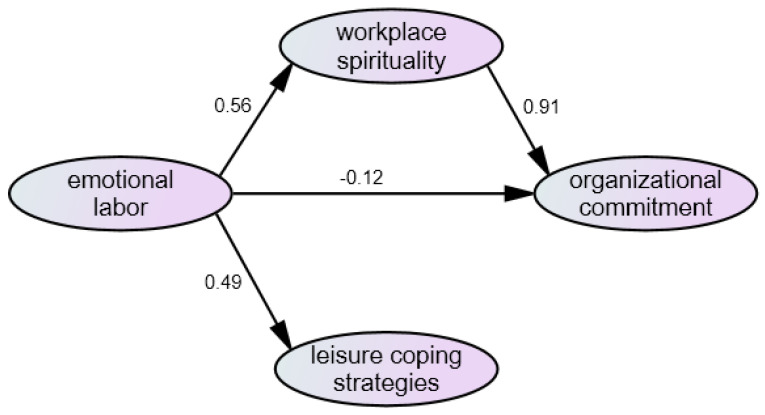
Model diagram of medical staffs’ Emotional Labor, workplace spirituality, Organizational Commitment, and Leisure Coping Strategies.

**Table 1 ijerph-19-06186-t001:** Sample characteristics.

Background Variables	Standard of Classification	Sample Size	Percentage (%)	Accumulative Percentage (%)
Gender	Male	45	15	15
	Female	255	85	100
Age	30 years old or below	61	20.3	20.3
	31–40 years old	106	35.3	55.7
	41–50 years old	87	29.0	84.7
	51 years old and above	46	15.3	100.0
Marital status	Married	182	60.7	60.7
	Unmarried	118	39.3	100.0
Educational level	High/vocational school (and below)	12	4.0	4.0
	University (including junior college)	221	73.7	77.7
	Graduate school (and above)	67	22.3	100.0
Place of residence	Northern area	30	10.0	10.0
	Central area	222	74.0	84.0
	Southern area	35	11.7	95.7
	Eastern area	13	4.3	100.0
Job title	Executive	56	18.7	18.7
	Non-executive	244	81.3	100.0
Years of service	Less than 2 years	3	1.0	1.0
	2–4 years (exclusive)	41	13.7	14.7
	4–6 years (exclusive)	87	29.0	43.7
	6–8 years (exclusive)	32	10.7	54.3
	8–10 years (exclusive)	15	5.0	59.3
	10 years or more	122	40.7	100.0
Do you often work overtime on “weekdays”?	Yes	84	28.0	28.0
	Sometimes	99	33.0	61.0
	No	117	39.0	100.0
Do you often work overtime on “holidays” (Saturdays, Sundays, or national holidays)?	Yes	73	24.3	24.3
	Sometimes	94	31.3	55.7
	No	133	44.3	100.0

**Table 2 ijerph-19-06186-t002:** Summary of the convergent validity and dimension reliability emotional labor.

Dimension	Index	Standardized Factor Loading	Cronbach’s a	C.R.	AVE
Surface performance	B2	0.81	0.71	0.81	0.68
	B3	0.85			
Deep performance	B6	0.77	0.80	0.81	0.68
	B7	0.88			
Emotional expression requirements	B13	0.84	0.76	0.79	0.65
	B14	0.78			
Emotional diversity	B16	0.87	0.88	0.86	0.68
	B17	0.79			
	B18	0.82			

Source: The analysis and organization of the convergent validity test of the Emotional Service Scale based on the data collected by this study.

**Table 3 ijerph-19-06186-t003:** Summary of the convergent validity and dimension reliability leisure coping strategies.

Dimension	Index	Standardized Factor Loading	Cronbach’s a	C.R.	AVE
Leisure companionship	C3	0.75	0.89	0.89	0.69
	C4	0.90			
	C5	0.84			
	C6	0.83			
Leisure palliative coping	C8	0.60	0.79	0.80	0.59
	C9	0.78			
	C10	0.90			
Leisure mood enhancement	C12	0.85	0.92	0.93	0.77
	C13	0.91			
	C14	0.89			
	C16	0.86			

Source: The analysis and organization of the convergent validity test of the Leisure Adjustment Strategy Scale based on the data collected by this study.

**Table 4 ijerph-19-06186-t004:** Summary of the convergent validity and dimension reliability workplace spirituality.

Dimension	Index	Standardized Factor Loading	Cronbach’s a	C.R.	AVE
A sense of work significance	D3	0.83	0.89	0.89	0.74
	D4	0.85			
	D5	0.90			
A sense of community	D8	0.90	0.88	0.88	0.79
	D9	0.88			
A sense of organizational commitment	D11	0.86	0.88	0.88	0.66
	D12	0.82			
	D13	0.85			
	D14	0.72			

**Table 5 ijerph-19-06186-t005:** Summary of the convergent validity and dimension reliability organizational commitment.

Dimension	Index	Standardized Factor Loading	Cronbach’s a	C.R.	AVE
Effort commitment	F1	0.77	0.79	0.79	0.65
	F2	0.85			
Retention commitment	F6	0.87	0.89	0.88	0.80
	F8	0.92			
Value commitment	F9	0.92	0.90	0.90	0.81
	F10	0.89			

Source: The analysis and organization of the convergent validity test of the Organizational Commitment Scale based on the data collected by this study.

**Table 6 ijerph-19-06186-t006:** Bootstrap correlation coefficient 95% confidence interval of Emotional Labor.

			Bias-Corrected			Percentile Method	
			Estimate	Lower Limit	Upper Limit	Lower Limit	Upper Limit
Surface performance	↔	Deep performance	0.81	0.70	0.90	0.70	0.91
Surface performance	↔	Emotional expression requirements	0.67	0.57	0.77	0.57	0.77
Surface performance	↔	Emotional diversity	0.47	0.33	0.61	0.31	0.59
Deep performance	↔	Emotional expression requirements	0.78	0.67	0.87	0.67	0.87
Deep performance	↔	Emotional diversity	0.60	0.47	0.72	0.46	0.72
Emotional expression requirements	↔	Emotional diversity	0.63	0.51	0.75	0.50	0.75

**Table 7 ijerph-19-06186-t007:** Bootstrap correlation coefficient 95% confidence interval of Leisure Coping Strategies.

			Bias-Corrected			Percentile Method	
			Estimate	Lower Limit	Upper Limit	Lower Limit	Upper Limit
Leisure companionship	↔	Leisure palliative coping	0.72	0.62	0.81	0.61	0.81
Leisure companionship	↔	Leisure mood enhancement	0.69	0.60	0.77	0.59	0.77
Leisure palliative coping	↔	Leisure mood enhancement	0.89	0.81	0.97	0.82	0.97

**Table 8 ijerph-19-06186-t008:** Bootstrap correlation coefficient 95% confidence interval of workplace spirituality.

			Bias-Corrected			Percentile Method	
			Estimate	Lower Limit	Upper Limit	Lower Limit	Upper Limit
A sense of work significance	↔	A sense of community	0.64	0.55	0.74	0.54	0.73
A sense of work significance	↔	A sense of organizational commitment	0.63	0.53	0.71	0.52	0.70
A sense of community	↔	A sense of organizational commitment	0.90	0.85	0.96	0.84	0.96

**Table 9 ijerph-19-06186-t009:** Bootstrap correlation coefficient 95% confidence interval of Organizational Commitment.

			Bias-Corrected			Percentile Method	
			Estimate	Lower Limit	Upper Limit	Lower Limit	Upper Limit
Effort commitment	↔	Retention commitment	0.70	0.61	0.78	0.61	0.78
Effort commitment	↔	Value commitment	0.73	0.64	0.81	0.64	0.81
Retention commitment	↔	Value commitment	0.89	0.84	0.94	0.84	0.94

**Table 10 ijerph-19-06186-t010:** Summary of the mediating effect.

	Estimate	95% Confidence Interval		
Indirect Effect		BC/PC *p*-Value	BC	PC
Emotional labor→workplace spirituality→organizational commitment	0.508	0.001/0.002	0.357–0.742	0.341–0.726
Direct effect				
Emotional labor→workplace spirituality	0.559	0.002/0.002	0.393–0.714	0.391–0.713
Emotional labor→organizational commitment	−0.117	0.060/0.055	−0.255–0.004	0.003–0.055
Workplace spirituality→organizational commitment	0.910	0.002/0.002	0.805–1.057	0.803–1.046
Total effect				
Emotional labor→organizational commitment	0.559	0.002/0.002	0.229–0.555	0.225–0.549

**Table 11 ijerph-19-06186-t011:** Analysis on the research model overall goodness of fit.

Fit Index	Allowable Standard	Model Corrected	Mode Fit Judgment
χ^2^ (Chi-square)	The smaller, the better	1115.51	
Ratio of χ^2^ to degrees of freedom	<3	2.05	Conforming
GFI	>0.80	0.83	Conforming
AGFI	>0.80	0.80	Conforming
RMSEA	<0.08	0.06	Conforming
CFI	>0.80	0.93	Conforming
PCFI	>0.50	0.85	Conforming

Source: The analysis and organization of the research model overall goodness of fit based on the data collected by this study.

**Table 12 ijerph-19-06186-t012:** Summary of the research hypotheses and test results.

Research Hypothesis	Path Value	Test Result
**Hypothesis 1 (H1).***Emotional labor has a significant impact on workplace spirituality*.	0.56	Established
**Hypothesis 2 (H2).***workplace spirituality has a significant impact on organizational commitment*.	0.91	Established
**Hypothesis 3 (H3).***Emotional labor has a significant impact on organizational commitment*.	−0.12	Established
**Hypothesis 4 (H4).***Emotional labor has a significant impact on leisure coping strategies*.	0.49	Established
**Hypothesis 5 (H5).***workplace spirituality has a significant mediating effect between emotional labor and organizational commitment*.		Not established

Source: The organization of the research hypotheses and test results of the path analysis were based on the data collected by this study.

## Data Availability

The data that support the findings of this study are available from the corresponding author upon reasonable request.

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
