# Peer review of "The Relationship between Medical Staff’s Emotional Labor, Leisure Coping Strategies, Workplace Spirituality, and Organizational Commitment during the COVID-19 Pandemic"

_ijerph, 2022, doi:10.3390/ijerph19106186_

Round 1

Reviewer 1 Report

While the topic/issue of emotional labor and the impact on workers during and in the wake of the pandemic is an important topic that is relevant to the journal, the current paper requires significant improvement before being considered for publication in the journal.  The current paper reads more like an exploratory study that is part of a larger efforts and thus lacks clear focus.  This makes the overall contribution of the current research very limited. It is likely that there are unique findings within the larger research project, but the current findings about the connection between coping strategies, emotional labor and organizational commitment are well-documented by existing research.  Thus a stronger case on the important and unique contribution of the current research to existing work must be made in order to this to be suitable for publication.  I would also stronger suggest that the authors provide a clearer rationale for each of the 4 stated (and 5 in actuality) focus areas of the paper. The introduction treats each of these variables as individual factors but what is being tested is a model that is not clearly explained and supported by a complete review of existing research that uses these variables in one specific model.  If the core focus is on emotional labor as the title would indicate, then the subsequent models being tested should be more clearly framed as a model for better understanding emotional label which is not the case within the current manuscript.  The inclusion of workplace spirituality is very disconnected from the other factors especially in the content of "leisure coping strategies". There is also confusion between organizational commitment and action by leadership which are two very different concepts both in terms of measuring and impact on key outcomes. It is also a concern that some factors are labeled as "leisure activities", "so-called workplace spirituality" and "so-called coping strategies" within the paper.  This is confusing in terms of the overall contribution of the work is the challenge, or to validate or to merely describe these "so-called" factors.  This labeling is very confusing to the readers.  The entire review of spirituality also needs a clearer focus as personal spirituality is mixed together with workplace spirituality, organizational spirituality and spirituality perception which are all different concepts.  The results section needs significant improvement.  More description about the sample (e.g. recruitment method, response rates by key demographics) are needed.  The measures needed to be combined into concept scales with alpha coefficients clearly provided. The lengthy correlation tables are not critical to present in such detail - a summary table or outcomes and scale coefficients would be a more effective presentation of these findings. Figure 2 is very challenging to understand and it appears that individual items are used in the regression analysis which is not appropriate for these data.  A test of mediation and moderation is included but it is not clear from the introduction why these two tests are selection and what specifically is driving these specific analyses.  As a result, the 5 key findings that are identified are well documented in prior research so the contribution to existing work is not clearly shown in the current version of this paper.  One suggestion would be for the authors to examine the extensive research on "work as calling" which may provide a more coherent framework for the individual variables that are within their database and provide more direction and explanation for the regression analyses that are presented in the paper.  Thus the impact on current and future work would become clearer as contribution to the exiting research on work as calling particular within the content of medical and healthcare professionals throughout the recent pandemic.

Author Response

Dear Reviewer,

Thank you for the constructive suggestions and comments on our manuscript (ID: ijerph-1680935). The suggestions and comments are helpful for improving the manuscript. We are submitting the revised version of the manuscript with our responses to the suggestions and comments by the reviewer. Many thanks for your guidance.

Our responses to each suggestion and comment are as follows, and they are presented in blue texts with a grey background color in the revised manuscript.

Reviewer 2 Report

Comments to the Author
The article by Yuan et al. describes the results of cross-sectional surveys done in Taiwan to figure out the relationship between workplace spirituality, medical staffs' emotional labor, and organizational commitment.

1.    The article, the introduction and the results, in particular, are quite lengthy, redundant, and hard to grasp what the authors are trying to say. I recommend significantly shortening the introduction, combining tables, or putting unnecessary data as supplementary data. 

2.    (Introduction) Data regarding COVID-19 incidence/mortality is rapidly changing, and I do not think it is beneficial to include the quantitative data at a particular time point. I would omit the numbers provided in introduction 1.1.

3.    (Methods) The authors failed to define who ‘medical staffs’ are. Did you include MDs, nurses, nurse aides, and what exactly their job categories were? The authors divided the participants into ‘Executive’ and ‘Non-executive,’ which does not make sense as well.

4.    (Methods) The authors mentioned that they distributed the questionnaire to 360 people but did not mention where they distributed it (What hospital? Which website? What platform, google form or Qualtrics?). 

5.    (Methods) Please include the questionnaire itself as a part of supplementary data.

6.    (Tables) Very hard to read. Please describe abbreviations as approproiate in table legends for every table. 

7.    (Discussion and conclusions) The authors restated previously proposed hypotheses and failed to address new findings. Discussions are based on authors’ opinions or empiric thoughts and are not backed up by evidence.  

Author Response

(The authors gave the same response as above.)

Reviewer 3 Report

Dear colleagues, I hope this message find you well.

Thank you for giving me the opportunity of reading the work “The Relationship between Medical Staff’s Emotional Labor, Leisure Coping Strategies, Workplace Spirituality, and Organizational Commitment during the COVID-19 Pandemic, it has been a very big pleasure to collaborate reviewing this manuscript. The topic of this paper is very interesting and it seems necessary to delve it. However, there are several questions to improve before to publish it. I would suggest some changes: 

Title and abstract

  • As a suggestion, if the authors consider it, some context could be added at the beginning of the abstract (1 or 2 lines).

Introduction

  • Dear colleagues, the structure of the introduction is clear, congratulations. However, I recommend to reduce the background in order to facilitating the reading.
  • Secondly, when you explain the signs of psychological distress and mental health as a result of COVID-19, I consider necessary to add more data and references. I recommend you to add this paper recently published (https://doi.org/10.3390/ijerph18147422), which proposes COVID-19 pandemic as a PTSD.
  • Moreover, introducing more studies could be interesting in order to support better how COVID-19 has psychologically affected health-care professionals. I suggest:

https://doi.org/10.3390/jcm10184077

https://doi.org/10.1016/S2215-0366(20)30307-2

10.1016/S2215-0366(20)30089-4

https://doi.org/10.1016/j.jaac.2020.08.466

Method

  • 5. Research Tools: Sample items should be added in the main text.

Results

  • Your research and results are fantastically developed and explained, congratulations.

Discussion

  • In my humble opinion, it could be useful to describe in more detail the practical and theoretical implications of this research. Considering this contribution as a part of a special issue, it would be useful they contextualize better the contribution within the framework of the issue explaining why the contribution is useful and enrich the impact.

Conclusions

  • Nothing to add. Good job.

Author Response

(The authors gave the same response as above.)

Reviewer 4 Report

Thank you for the opportunity to review the article: The Relationship between Medical Staff’s Emotional Labor, Leisure Coping Strategies, Workplace Spirituality, and Organizational Commitment during the COVID-19 Pandemic.

I appreciate the choice of topic and the effort and commitment of the Authors in the preparation of the work, but in my opinion, the article requires rewriting and organizing the structure – in accordance with the guidelines for authors. (Introduction, Materials and Methods, Results, Discussion, Conclusions)

My suggestions for Authors:

  1. Introduction

The section contains more than 4 pages - it could be shortened a bit.

  1. Methods

In this section, you can limit yourself to hypotheses only (not necessarily include research problems).

I also suggest completing:

Participants of the study - description of the study group (what exactly medical personnel) criteria for inclusion and exclusion in the study.

The Course of the Study - the place where the research was conducted, where the electronic survey was posted.

  1. Results and Discussion

In this section, the Authors use only laconic descriptions and refer the reader to tables. I would suggest that you expand on these descriptions a bit and include at least brief descriptions of the most important results – above each table. (e.g. Point 3.1 has only one sentence).

The Authors combined the results with a discussion of what is acceptable but taking into account how rich the material they obtained, it is worth dividing them into two sections. I also suggest that section 5 Suggestions (Implications, limitations, future review directions) be included at the end of the discussion.

Please include explanations of the abbreviations under each table.

  1. Conclusions–I propose to put the conclusions in a descriptive way, not in points.
  2. Suggestions -this part could be included at the end of the discussion.

Author Response

(The authors gave the same response as above.)

Round 2

Reviewer 1 Report

  • The research problems are redundant with the research hypotheses especially since the follow the hypothesis within the paper. I would strongly suggest deleting the research problems or moving them to the introduction section of the paper (before the Methods Section).
  • Use of the phrase “no number 1” or “The results showed that 1 did not appear” is confusing for the reader so an alternative wording in strongly recommended here throughout the results section.
  • Hypothesis #5 (“Workplace spirituality has a significant mediating effect between emotional labor and organizational commitment.”) is not depicted in Figure 1 and should somehow be added to that figure.
  • Given this result - “Since the p value was not significant, there was no mediating effect, as 428 shown in Table 10” – it should be explicitly stated that “hypothesis 5 is not supported”.
  • Figure 2 is extremely difficult to read and understand. I would strongly recommend deleting the items since they are already defined in detail within the test and retaining the concepts only as a clearer illustration of the findings.
  • The conclusions states that “Finally, this study found that the mediating effect of medical staffs’ workplace spirituality 606 between emotional labor and organizational commitment was not significant. A suggestion is to add 1-2 sentences to identity the implications/suggestions for future research.

Author Response

Dear Reviewer,

Thank you again for the constructive suggestions and comments on our manuscript (ID: ijerph-1680935). The suggestions and comments are helpful for improving the manuscript. We are submitting the revised version of the manuscript with our responses to the suggestions and comments by the reviewer. Many thanks for your guidance.

Our responses to each suggestion and comment are as follows, and they are presented in blue texts with a grey background color in the revised manuscript.

Reviewer 2 Report

The authors appropriately addressed my initial concerns.

Author Response

(The authors gave the same response as above.)

Reviewer 4 Report

The Authors took into account all suggestions and the paper has been improved.

I suggest only adding legends under the tables where abbreviations are (tables 2, 3, 5, 11).

Author Response

(The authors gave the same response as above.)
